# A Mechanistic Pharmacodynamic Modeling Framework for the Assessment and Optimization of Proteolysis Targeting Chimeras (PROTACs)

**DOI:** 10.3390/pharmaceutics15010195

**Published:** 2023-01-05

**Authors:** Robin Thomas Ulrich Haid, Andreas Reichel

**Affiliations:** 1DMPK Modeling and Simulation, Drug Metabolism and Pharmacokinetics, Preclinical Development, Bayer AG, Müllerstraße 178, 13353 Berlin, Germany; 2Biopharmacy, Institute of Pharmaceutical Sciences, Department of Chemistry and Applied Biosciences, ETH Zurich, Vladimir-Prelog-Weg 4, 8093 Zurich, Switzerland

**Keywords:** PK/PD, targeted protein degradation, proteolysis targeting chimera, PROTAC, event-driven pharmacology, hook effect, model-informed drug discovery, translational pharmacology, experimental design

## Abstract

The field of targeted protein degradation is growing exponentially. Yet, there is an unmet need for pharmacokinetic/pharmacodynamic models that provide mechanistic insights, while also being practically useful in a drug discovery setting. Therefore, we have developed a comprehensive modeling framework which can be applied to experimental data from routine projects to: (1) assess PROTACs based on accurate degradation metrics, (2) guide compound optimization of the most critical parameters, and (3) link degradation to downstream pharmacodynamic effects. The presented framework contains a number of first-time features: (1) a mechanistic model to fit the hook effect in the PROTAC concentration-degradation profile, (2) quantification of the role of target occupancy in the PROTAC mechanism of action and (3) deconvolution of the effects of target degradation and target inhibition by PROTACs on the overall pharmacodynamic response. To illustrate applicability and to build confidence, we have employed these three models to analyze exemplary data on various compounds from different projects and targets. The presented framework allows researchers to tailor their experimental work and to arrive at a better understanding of their results, ultimately leading to more successful PROTAC discovery. While the focus here lies on in vitro pharmacology experiments, key implications for in vivo studies are also discussed.

## 1. Introduction

Targeted protein degradation (TPD) is a rapidly growing novel therapeutic approach. It utilizes the cell’s ubiquitin proteasome system (UPS) to specifically degrade and thus completely remove target proteins instead of inhibiting their function as most drugs do [1]. The TPD approach employs heterobifunctional drug molecules, so-called proteolysis targeting chimeras (PROTACs), that simultaneously bind to a target protein and an E3 ligase to form a ternary complex. The target protein, also referred to as protein of interest (POI), is then polyubiquitinated by the selected E3 ligase which in turn triggers its degradation by the proteasome [2].

In recent years, the field has been growing exponentially [3,4] with several PROTACs having entered clinical trial status [5]. Yet, there are only few publications on PK/PD models that quantitatively link pharmacokinetics (PK) with their unique mechanism of action (MOA) and downstream pharmacodynamic (PD) effects. Of those, the ones that are best established focus on describing ternary complex formation mathematically [6,7]. In contrast, quantitative modeling of the so-called *event-driven* MOA of PROTACs, as opposed to the *occupancy-driven* MOA of classical inhibitors, is only in its infancy. Recently, Bartlett and Gilbert generated a theoretical PK/PD model [8] which however was only sparsely applied to experimental data. We have therefore set out to develop a mechanistic PK/PD modeling framework which is directly applicable to data as generated in early PROTAC drug discovery projects.

This framework provides practical guidance for project teams engaged in the quantitative assessment and rational optimization of PROTACs. We have used published and in-house experimental data to validate our models and to demonstrate their practical utility. Moreover, we only use parameters that are readily accessible to project teams. To ensure its translational value, our framework is built on the *three pillars of clinical success* concept which has become a hallmark of model-informed drug discovery and development (MID3) [9,10,11]. The concept was recently applied to PROTACs [12] and expanded to *four pillars of translational pharmacology* where successful drug action requires the drug to (I) be sufficiently exposed to its target(s), (II) engage its target(s), and (III) modulate the activity of its target(s), which in turn (IV) elicits the desired downstream pharmacodynamic effect(s) (see Figure 1). In the present article, we focus exclusively on pillars II–IV (PD) since, in the early phases of a project, assessment and optimization of compounds is based mainly on in vitro studies. For considerations on pillar I, i.e., the pharmacokinetics (PK) of PROTACs, please refer to recent publications by Pike et al. [13] and Cantrill et al. [14].

The presented modeling framework has three principal areas of application which have not been addressed in a satisfactory manner before. First, it allows to estimate key degradation parameters such as $D_{\max}$ (the maximal extent of degradation) and $DC_{50}$ (the PROTAC concentration that produces half-maximal degradation) even if the experimental system is not at steady state. These accurate estimates can then be used to establish more solid structure-activity relationships by avoiding mis-ranking of compounds. Furthermore, they will also inform the selection of the most promising compounds for the next experimental testing steps. Our model thus is a powerful tool for optimizing the design of in vitro experiments, and to maximize their value while minimizing experimental efforts.

The second application area of the modeling framework is geared towards the rational optimization of PROTACs with regard to target protein degradation. There, the model allows to simulate the impact of the PROTAC’s binding affinities for POI and E3 ligase on degradation. Thus, it provides target values for the optimization of these parameters and informs the design of follow-up compounds. In addition, the model gives key insight into what levels of target occupancy and catalytic efficiency of the PROTAC have been achieved in the selected cellular system.

The third application area of the modeling framework lies in quantitatively linking target protein degradation to downstream pharmacodynamic effects. This part of the model furthermore allows to delineate the relative contributions to the overall PD effect of target degradation and target inhibition. Target inhibition by the PROTAC occurs if the POI binding moiety derives from an inhibitor as very often is the case. This composite MOA has important implications for the downstream translation of the hook effect.

The presented modeling framework is not only a very practical toolbox for project teams, but it also provides important mechanistic insights that were not accessible before. For example, it highlights how important target occupancy is despite PROTAC action often being classified as event-driven. Moreover, it allows to quantitatively delineate multiple MOAs. Our modeling framework is novel as it contains several features for the first-time and readily enables model-informed assessment and optimization in early PROTAC drug discovery. While the present article focuses on in vitro degradation and PD data, an extension of the modeling framework towards in vivo will be subject of a separate publication.

## 2. Materials and Methods

### 2.1. Modeling Approach

We aimed to generate a PK/PD modeling framework that is directly applicable for the typical questions in early PROTAC drug discovery. Moreover, we wanted it to also reflect the PROTAC MOA and thus we combined the fit-for-purpose approach with experimentally driven learn-and-confirm cycles. Ultimately, we set out to derive a clear rational guidance for project teams with translational value spanning from target degradation to the desired pharmacological effects. The model was built and validated using in-house and literature data, which were extracted from the respective publications using version 4.5 of the *WebPlotDigitizer* tool [15]. Data analysis was conducted in *R* version 4.1.2 [16] using the integrated development environment *R-Studio* version 2022.07.2+576 [17]. Statistical significance was assessed based on t-tests conducted on a 95% confidence level.

### 2.2. Pillar II—Target Engagement

In contrast to conventional small molecule drugs, PROTACs are bispecific that is they have two different targets: the target protein (POI), which is to be degraded, and the E3 ligase (E3) that marks the target protein for UPS-mediated degradation. The PROTAC binds to both of its targets one by one, which initially yields a binary complex (POI_PROTAC or PROTAC_E3) and ultimately affords the ternary complex (POI_PROTAC_E3). As degradation completely relies on ternary complex formation, target engagement (TE) is defined as the fraction of total target protein (P) that is in a ternary complex (TC):(1)$$TE∶=\frac{TC}{P}\;.$$

We describe complex formation with the respective thermodynamic equilibrium constants according to a rapid equilibrium model [18]. The propensity to form binary complexes is given by the complex dissociation constants ($K_{D,P}$ and $K_{D,E}$), while the cooperativity factor (α) links binary complex affinities to ternary complex affinity. With regard to concentrations, the total (i.e., the sum of unbound and bound to PROTAC) levels of targeted E3 ligase are assumed to be constant over time ($E_{0}$). In vitro, the same should hold true for the unbound PROTAC concentrations (C). Utilizing an expression for the relationship between drug exposure and ternary complex formation derived by Han [6] (Equation (2-1) in Ref. [6]), target engagement can be written as:(2)$$TE=\frac{1}{2}\times \left(1+\frac{E_{0}}{P}+\frac{\left(K_{D,P}+C\right)\times \left(K_{D,E}+C\right)}{P\times \alpha \times C}-\sqrt{{\left(1+\frac{E_{0}}{P}+\frac{\left(K_{D,P}+C\right)\times \left(K_{D,E}+C\right)}{P\times \alpha \times C}\right)}^{2}-4\times \frac{E_{0}}{P}}\right)\;.$$

If we assume that the cell’s degradation machinery, including the targeted E3 ligase, does not get saturated by the target protein, this becomes (see Appendix B for derivation):(3)$$TE=\frac{\alpha \times E_{0}}{\alpha \times E_{0}+K_{D,P}+K_{D,E}+K_{D,P}\times K_{D,E}\times \frac{1}{C}+C\;}.$$

The rationale behind the above assumption is to write TE as an expression that does not depend on target protein levels. Thus, $P_{0}$ does not show up in Equation (3) and, therefore, also does not have to be determined experimentally (see Section 4 for implications).

### 2.3. Pillar III—Target Modulation

Target modulation (TM) quantifies how much baseline target activity has been altered. For PROTACs, we define it as the relative difference between baseline levels of target protein ($P_{0}$) and levels of active protein ($P_{a}$) remaining after drug application:(4)$$TM∶=\frac{P_{0}-P_{a}}{P_{0}}=1-\frac{P_{a}}{P_{0}}\;.$$

The target binding moieties of PROTACs are commonly based on small molecule inhibitors. Therefore, besides degrading the target protein, PROTACs also inhibit what activity remains. Thus, it makes sense to think of target modulation in terms of the fraction of the target that is degraded (D) and the fraction of the (remaining) target that is inhibited (I). The levels of active protein remaining after drug application are then given by the fraction of target protein that is neither degraded nor inhibited:(5)$$\frac{P_{a}}{P_{0}}=\left(1-D\right)\times \left(1-I\right)\;.$$

Inserting this expression into Equation (4) allows to link target modulation to degradation and inhibition:(6)TM=1−(1−D)×(1−I)=D+I−D×I .

### 2.4. Pillar III.a—Target Degradation

The extent of target degradation (D) is defined in terms of the amount of target protein remaining relative to baseline:(7)$$D∶=1-\frac{P}{P_{0}}\;.$$

The kinetics of PROTAC-mediated degradation are commonly described by an indirect response model as was pointed out in a review by Bartlett and Gilbert [19] (Equation (3) in Ref. [19]). The change in relative target levels over time (t) can then be written as:(8)$$\frac{d}{dt}\left(\frac{P}{P_{0}}\right)=k_{\deg,P}-k_{\deg,P}\times \frac{P}{P_{0}}-k_{\mathrm{cat}}\times TE\times \frac{P}{P_{0}}\;.$$

There are two competing processes at play here. On the one hand, there is the cell’s intrinsic protein turnover, which is governed by a first-order rate constant ($k_{\deg,P}$). On the other hand, there is the degradation that is catalyzed by the PROTAC, which has its own rate constant associated with it ($k_{\mathrm{cat}}$). Inspecting Equation (3) reveals that, in vitro, target engagement does not change over time, which allows to readily solve the above differential equation:(9)$$\frac{P}{P_{0}}=1-\frac{k_{\mathrm{cat}}\times TE}{k_{\deg,P}+k_{\mathrm{cat}}\times TE}\times \left(1-\exp\left(-\left(k_{\deg,P}+k_{\mathrm{cat}}\times TE\right)\times t\right)\right)\;.$$

Applying the definition for target degradation (Equation (7)) and expressing target protein turnover in terms of its half-life ($t_{½,P}$) gives:(10)$$D=\frac{k_{\mathrm{cat}}\times t_{½,P}\times TE}{\ln\left(2\right)+k_{\mathrm{cat}}\times t_{½,P}\times TE}\times \left(1-\exp\left(-\left(\frac{\ln\left(2\right)}{t_{½,P}}+k_{\mathrm{cat}}\times TE\right)\times t\right)\right)\;.$$

Finally, inserting the expression for target engagement (Equation (3)) yields what we will call the $k_{cat}$
*model*:(11)$$\begin{array}{c} D\left(C,t\right)=D_{\mathrm{ss}}\left(C\right)\times \left(1-\exp\left(-\frac{\ln\left(2\right)\times t}{t_{½,P}\times \left(1-D_{\mathrm{ss}}\left(C\right)\right)}\right)\right)\;,\\ \mathrm{where}:\;D_{\mathrm{ss}}\left(C\right)=\frac{\alpha \times E_{0}\times k_{\mathrm{cat}}\times t_{½,P}\times \frac{1}{\ln\left(2\right)}}{\alpha \times E_{0}\times k_{\mathrm{cat}}\times t_{½,P}\times \frac{1}{\ln\left(2\right)}+\alpha \times E_{0}+\frac{\;K_{D,P}\times K_{D,E}}{C}+K_{D,P}+K_{D,E}+C}\;. \end{array}$$

The term $D_{\mathrm{ss}}$ in the above equation refers to the degradation at steady state that is the extent of degradation if incubation went on indefinitely.

Lumping together the parameters in the expression for $D_{\mathrm{ss}}$ above, gives what we will call the *hook model*:(12)$$D\left(C\right)=D_{\max}\times \frac{DC_{50}{}^{2}+DC_{\max}{}^{2}-2\times DC_{50}\times DC_{\max}}{DC_{50}{}^{2}+DC_{\max}{}^{2}-4\times DC_{50}\times DC_{\max}+DC_{50}\times C+\frac{DC_{50}\times DC_{\max}{}^{2}}{C}}\;,$$
where $D_{\max}$ is the maximal extent of degradation, $DC_{50}$ the drug concentration that produces half-maximal degradation and $DC_{\max}$ the concentration that gives maximal degradation (see Appendix C for how the parameters are related). Both, the $k_{cat}$ model and the hook model deal with target degradation. While the $k_{cat}$ model can be used to predict target degradation from biochemical parameters (a priori), the hook model is used to fit experimental concentration-degradation profiles (a posteriori). For PROTAC concentrations that are much smaller than the concentration of maximal degradation ($C\ll DC_{\max}$), Equation (12) can be approximated by the hyperbolic $E_{\max}$ model (see Appendix C for derivation):(13)$$E\left(C\right)=E_{\max}\times \frac{C}{C+EC_{50}}\;,$$
with the effect being target degradation (E=D, $E_{\max}=D_{\max}$ and $EC_{50}=DC_{50}$).

Since the extent of degradation depends on incubation time (see Equation (11)), the descriptive parameters ($D_{\max}$ and $DC_{50}$, but not $DC_{\max}$) do so as well. The hook model can be extended to account for this time dependence:(14)$$\begin{array}{c} D\left(C,t\right)=D_{\mathrm{ss}}\left(C\right)\times \left(1-\exp\left(-\frac{\ln\left(2\right)\times t}{t_{½,P}\times \left(1-D_{\mathrm{ss}}\left(C\right)\right)}\right)\right)\;,\\ \mathrm{where}:\;D_{\mathrm{ss}}\left(C\right)=D_{\max}\times \frac{DC_{50}{}^{2}+DC_{\max}{}^{2}-2\times DC_{50}\times DC_{\max}}{DC_{50}{}^{2}+DC_{\max}{}^{2}-4\times DC_{50}\times DC_{\max}+DC_{50}\times C+\frac{DC_{50}\times DC_{\max}{}^{2}}{C}}\;. \end{array}$$

The parameters ($D_{\max}$, $DC_{50}$ and $DC_{\max}$) in the above equation refer to the respective values at steady state.

Finally, degradation can also be linked directly to target engagement. Using Equation (3), the expression for degradation at steady state from Equation (11) can be rewritten in terms of TE:(15)$$\begin{array}{c} D_{\mathrm{ss}}\left(TE\right)=\frac{TE}{TE_{50}+TE}\;,\\ \mathrm{where}:\;TE_{50}∶=\frac{\ln\left(2\right)}{k_{\mathrm{cat}}\times t_{½,P}}=\frac{k_{\deg,P}}{k_{\mathrm{cat}}}\;. \end{array}$$

Thus, the ratio between baseline protein turnover and the PROTAC-mediated degradation rate constant gives the target engagement required for 50% protein degradation at steady state ($TE_{50}$). We will refer to this parameter as the PROTAC’s catalytic efficiency in a given cell type.

### 2.5. Pillar III.b—Target Inhibition

Inhibition (I) can be defined in terms of the fraction of the remaining (i.e., non-degraded) target protein (P) that is still active ($P_{a}$).
(16)$$I∶=1-\frac{P_{a}}{P}\;.$$

If we assume that binding to the PROTAC completely shuts down target protein activity, this fraction is equivalent to the target protein’s unbound fraction, given by (Equation (1–7) in Ref. [6]):(17)$$\frac{P_{a}}{P}=\frac{P_{u}}{P}=\frac{K_{D,P}\times \left(1-TE\right)}{K_{D,P}+C}\;.$$

Furthermore, we will assume that only a small fraction of the target protein is part of a ternary complex at any point in time. This appears reasonable when considering that PROTACs act as catalysts that is in a substoichiometric manner. Thus, target engagement is low (i.e., TE≪1) and inhibition is mainly driven by the formation of binary complexes between the target protein and the PROTAC:(18)$$\frac{P_{a}}{P}=\frac{K_{D,P}}{K_{D,P}+C}\;.$$

Inserting this expression into Equation (16) allows to write inhibition as:(19)$$I=\frac{C}{K_{D,P}+C}\;.$$

### 2.6. Pillar IV—Pharmacodynamic Effects

Protein degradation usually aims at interfering with a biological pathway to produce a certain downstream pharmacodynamic response. The target protein might, for example, lead to the release of a certain signaling molecule and the PROTAC then shuts that signal down. Assuming this process is rapid compared to degradation, we propose a direct $E_{\max}$ model to link target modulation to the pharmacodynamic response (PD):(20)$$\begin{array}{c} PD=PD_{\min}+\left(1-PD_{\min}\right)\times \frac{{\left(1-TM\right)}^{n}\times \left(1-P_{50}{}^{n}\right)}{P_{50}{}^{n}+{\left(1-TM\right)}^{n}\times \left(1-2\times P_{50}{}^{n}\right)}\;,\\ \mathrm{where}:\;P_{50}<\sqrt[{n}]{\frac{1}{2}}\;. \end{array}$$

Note that PD in the above equation refers to the relative pharmacodynamic response, normalized to its baseline value.

According to this PD
*model* (see Appendix D for derivation), the pharmacodynamic response decreases with increasing target modulation. However, there might also be a certain extent of response that is independent of the target protein and thus not affected by PROTAC treatment ($PD_{\min}$). Furthermore, there might be saturation of the pharmacodynamic response with regard to the target protein. This saturation effect is expressed by the target protein levels, where the target protein dependent pharmacodynamic response is reduced to half of its baseline value ($P_{50}$). Finally, there might be the need to include an empirical Hill-coefficient (n), especially when dealing with more downstream effects.

As the pharmacodynamic response depends on target modulation, it is affected by both target degradation as well as target inhibition. While target degradation can be measured experimentally, target inhibition by the PROTAC cannot be quantified that easily. Note that target inhibition in this context refers strictly to the fractional occupancy of the target protein rather than a surrogate marker. However, what can be measured instead is the effect of the corresponding non-degrading inhibitor on the pharmacodynamic response. Therefore, we rewrite (see Appendix D for derivation) Equation (19) in terms of the PROTAC concentration, where the pharmacodynamic response is reduced to half of its baseline value by inhibition alone ($IC_{50}$):(21)$$I=\frac{C\times \left(1-P_{50}\right)}{C\times \left(1-P_{50}\right)+IC_{50}\times P_{50}}\;.$$

### 2.7. Practical Application

Together, the three models derived above constitute a framework which links target engagement to target modulation and pharmacodynamic response. This then allows to integrate experimental results from different assays to arrive at a wholistic understanding of how these input parameters control the steps along the pillars of translational pharmacology (see Figure 2). Thereby, the following key questions can be addressed in a systematic manner:

How much degradation is there?   →  *hook model*How to increase the extent of degradation?   →  $k_{cat}$
*model*How much degradation is necessary?   →  *PD model*

How these questions can be answered with the presented framework is demonstrated in the following sections using data from various sources as illustrative examples (see Section 3 for the details and Section 5 for an overview).

## 3. Results

### 3.1. Assessing PROTACs as Degraders

#### 3.1.1. Capturing the Hook Effect

Already during the lead generation phase, it becomes necessary to analyze experimental data on PROTAC-mediated protein degradation with the help of mathematical models. Such models then allow to compare and rank different compounds with regard to their degradation parameters (e.g., $D_{\max}$ and $DC_{50}$). Currently, the PROTAC concentration-degradation relationship is most commonly described using the $E_{\max}$ model [20]. However, that model has often failed to capture the loss of degradation observed at higher PROTAC concentrations, as in the case of the compounds from Zorba et al. [21]. The hook model, in contrast, fitted that data well (see Figure 3a, Appendix A). Notably, the two models agreed almost perfectly at the lower concentrations, illustrating that the hook model represents a valid extension of the $E_{\max}$ model.

Moreover, the hook model accurately estimated $D_{\max}$, whereas the values suggested by the $E_{\max}$ model were significantly larger than the experimentally observed data (see Figure 3b). Judging from Kendall’s correlation coefficient (τ), the $E_{\max}$ model (τ=0.79) also performed worse than the hook model (τ=0.97) when used for ranking compounds according to their $D_{\max}$. We therefore conclude that the hook model is better suited for describing data on PROTAC-mediated protein degradation than the standard $E_{\max}$ model.

In order to further elucidate the applicability domain of the hook model, we also fitted it to data on covalent PROTACs published by Gabizon et al. [22]. Again, the hook model was able to capture the entire concentration-degradation profile, both for reversible covalent and irreversible covalent compounds (see Figure 3c). In conclusion, the hook model is not limited to classical non-covalent PROTACs, but also provides added value when dealing with different kinds of covalent protein degraders.

#### 3.1.2. Impact of Incubation Time

When designing in vitro experiments, it is critical to consider that incubation time influences the PROTAC concentration-degradation relationship, with degradation increasing over time. This phenomenon can be illustrated with data published by Mares et al. [23], who determined the extent of RIPK2 degradation at two different time points (6 h and 24 h). We used the 6 h data together with a literature value for the protein’s half-life ($t_{½,P}=45\;h$) [24] to inform the extended hook model (Equation (14)). Subsequently, we applied that model to predict degradation after 24 h (see Figure 3d). This prediction was in good agreement with the measured values, demonstrating that the extended hook model allows to extrapolate with regard to incubation time.

Next, we compared the two observed concentration-degradation profiles to the one that was predicted for the steady state (see Table 1). As suggested by the model, the PROTAC concentration of maximal degradation ($DC_{\max}$) appeared to stay constant. The maximal extent of degradation ($D_{\max}$) and the concentration of half-maximal degradation ($DC_{50}$), on the other hand, were subject to change. As expected, $D_{\max}$ increased over time whereas $DC_{50}$ decreased. Notably, $D_{\max}$ was already reasonably close to its steady-state value after incubation for 24 h, while $DC_{50}$ was still off by a factor of 2. We thus conclude that of all parameters $DC_{50}$ is the most sensitive to incubation time.

### 3.2. Model-Informed Optimization of PROTACs

#### 3.2.1. Binding Affinities

Once an initial lead compound has been identified, its degrader properties might yet require further improvement. This was the case for *Cpd. A*, for example, which is part of a series of nine BTK PROTACs developed by Zorba et al. [21]. In such a scenario, our $k_{\mathrm{cat}}$ model can guide medicinal chemistry by predicting how much each of the binding affinities should be increased to get the desired degradation. First, to inform the model, the pivotal parameter $k_{\mathrm{cat}}$ was obtained by fitting the protein degradation data from the initial lead compound (*Cpd. A*). The resulting fit based on $k_{\mathrm{cat}}=4.6{\;h}^{-1}$ was in good agreement with the experimental data (see black graph in Figure 4a). Notably, only this one parameter ($k_{\mathrm{cat}}$) was obtained by fitting, while all other parameters had already been informed by orthogonal assays (see Appendix A). This $k_{\mathrm{cat}}$ value was then also applied for all further predictions.

The model was used to predict the binding affinities that would give an improved concentration-degradation profile (see orange graph in Figure 4a). When tested in the degradation assay, the optimized compound (*Cpd. I*), featuring the exact binding affinities proposed by the model (see Appendix A), also exhibited the desired concentration-degradation profile (see orange points in Figure 4a). Furthermore, the maximal extent of degradation ($D_{\max}$) was predicted also for the other degraders from that series based on their respective binding affinities. The predictions matched the experimental data across a wide range of values (see black points in Figure 4b).

These results demonstrate that most of the differences in degradation that were observed between compounds could be predicted from their binding affinities ($K_{D,P}$, $K_{D,E}$ and α). Importantly, all three of these parameters need to be considered, indicating that there is not just one key binding parameter.

#### 3.2.2. Physiological Parameters

Just as some PROTACs are more efficacious than others, there are also certain cell types that are particularly susceptible to protein degradation. To account for this kind of variability, the nine compounds from Zorba et al. [21] had not only been assessed in Ramos cells but also in THP-1 cells. We used the above model to anticipate the maximal extent of degradation ($D_{\max}$) in this second cell line, where E3 ligase expression levels ($E_{0}$) were 50% lower (see Appendix A). For all nine compounds, the predicted values matched the experimental data well (see red points in Figure 4b). Again, binding affinities determine which compounds perform best. The expression levels of E3 ligase, on the other hand, control how susceptible the individual cell lines are to target degradation in general. Since there was less E3 ligase in THP-1 cells than in Ramos cells, there was also less degradation in THP-1 cells.

As cell lines, like Ramos cells or THP-1 cells, might not be perfectly representative of the situation in vivo, Zorba et al. [21] also studied primary rat splenocytes. There, BTK half-life ($t_{½,P}$) was about four times greater than in Ramos cells (see Appendix A). Our model accurately predicted the maximal extent of degradation ($D_{\max}$) for the one compound that had been tested in the splenocytes (see blue point in Figure 4b). There was more degradation in the primary cells than in the cell lines because protein half-life is longer, i.e., target protein resynthesis is slower there. In conclusion, the $k_{\mathrm{cat}}$ model accurately predicts the maximal extent of degradation ($D_{\max}$) in different cellular systems based on target protein half-life and E3 ligase expression levels (see Figure 4b). The basal expression levels of the target protein ($P_{0}$), on the other hand, have only negligible effects on degradation (see Appendix B for derivation).

When optimizing PROTACs, care must be taken to ensure a fast enough onset of degradation. We therefore also predicted the time-course of degradation in Ramos cells using the $k_{\mathrm{cat}}$ model (see Figure 4c). Target protein levels followed the expected exponential decay towards a steady-state value, at the rate suggested by the $k_{\mathrm{cat}}$ model. Notably, even if that steady-state value is known, it still requires the half-life of the target protein ($t_{½,P}$) to also predict the rate of degradation. We thus concluded that target protein half-life is the most pivotal of the physiological parameters when it comes to modeling the action of PROTAC drugs.

While predicting degradation represents the primary PK/PD modeling objective during lead optimization, target engagement (TE) might be of mechanistic interest. Thus, we next calculated the maximum extent of TE achieved for the nine compounds in both cell lines (using Equation (3)) and related that value to $D_{\max}$ (see points in Figure 4d). There was a clear quantitative relationship between target engagement and protein degradation (see Figure 4d), although the extent of target engagement remained low for all PROTACs achieving a maximum of just under 6% (TE<6%). These results demonstrate that target occupancy is critical to the mechanism of action of PROTACs. The low TE, which may well be typical for PROTACs in general, might not be further increased by increasing PROTAC concentrations due to the hook effect. Since the extent of target engagement is low, it takes several sequential rounds of ternary complex formation and ubiquitination to get significant degradation. As a result, PROTAC-mediated degradation is a slow process (see Figure 4c) compared to target inhibition which operates at much higher target occupancies.

### 3.3. Deriving a Target Value for Degradation

#### 3.3.1. Linking Degradation to the Pharmacodynamic Response

While target degradation represents the primary readout for PROTACs, the downstream pharmacodynamic response is usually therapeutically more relevant. Mares et al., for example, not only measured degradation of RIPK2, but they also investigated what effects their compounds had on TNF-alpha response [23]. Besides the PROTACs, they also tested the corresponding non-degrading (nd) control compounds, which have the same inhibitory activity as the respective PROTACs.

Our PD model now allows to translate a target value for that therapeutically relevant effect back to the level of protein degradation. First, we used the TNF-alpha data from the lead compound (see black points in Figure 5a) to inform the model, considering both, degradation (see black points in Figure 5b) and the inhibitory potency of the control compound (see black points in Figure 5c). The fitted model suggested that independent of the PROTAC used, 13% of RIPK2 degradation (i.e., 87% remaining undegraded) would reduce TNF-alpha response to 50% ($P_{50}=87\%$).

Subsequently, a target profile was proposed on the level of the TNF-alpha response (see red graph in Figure 5a). The model was then used to predict, which degradation profile would be required to meet that target (see red graph in Figure 5b). When the optimized compound (*Cpd. Y*), carrying the $D_{\max}$ and $DC_{50}$ suggested by the model (see red points in Figure 5b) was tested in the TNF-alpha assay, it also had the desired potency (see red points in Figure 5a). This agreement between predictions and experimental observations validates the model.

Our model furthermore demonstrates that RIPK2 degradation is indeed the primary mechanism for reducing TNF-alpha response in the case of these compounds (see Figure 5d). Inhibition, on the other hand, plays only a minor role, as was predicted by the PD model.

#### 3.3.2. Interplay of Degradation and Inhibition

In the previous example, the contribution of inhibition to the overall pharmacodynamic effect was negligible. However, this need not be the case, as we demonstrate using in-house data on PROTACs, which were first assessed with regard to target degradation (see solid graphs in Figure 6a and Appendix A). Next, the PD model was fitted to their downstream pharmacodynamic response (see dashed graph in Figure 6a and Appendix A). While there was a clear hook effect visible on the level of protein degradation, no such phenomenon was observed for the downstream response. This was likely due to inhibition taking over from degradation as the main driver of target modulation at higher concentrations (see Figure 6b). Notably, our model suggests that the inhibitory activity of PROTACs should always compensate for the hook effect in protein levels (see Appendix E for derivation).

Furthermore, the contributions of degradation and inhibition to the overall pharmacodynamic response may also change over time. To showcase this effect, the time-course of degradation was predicted for an in-house PROTAC based on a literature value for protein half-life ($t_{½,P}\approx 24\;h$) [24] (see blue graph in Figure 6c). The PROTAC’s inhibitory potency, in turn, was informed using data on the corresponding non-degrading control compound ($IC_{50}=120\;\mathrm{nM}$). Moreover, previous experiments had shown that there was no target-independent pharmacodynamic response ($BM_{\min}=0$), no saturation of the pharmacodynamic response ($P_{50}=0.5$) and no empirical Hill coefficient (n=1). From these inputs, the time-course of the pharmacodynamic response was predicted using the PD model (see green graph in Figure 6c). At the earlier time points, target inhibition was the dominant driver behind the downstream effect, while degradation only kicked in later (see Figure 6d). Note, that the contribution of inhibition to the overall effect was relatively strong because a high PROTAC concentration (C=100 nM) was used.

## 4. Discussion

### 4.1. The Hook Model

#### 4.1.1. Advancing Current Best Practices

It has been widely accepted that the hook effect constitutes a characteristic feature of the PROTAC mechanism of action [25,26]. Yet, when it comes to modeling concentration-degradation profiles, researchers still default to the $E_{\max}$ model which, as it does not account for it [27,28,29,30], carries several disadvantages.

First, concentrations that are greater than the concentration of maximal degradation ($DC_{\max}$) must often be excluded from the analysis to make the $E_{\max}$ model converge [31]. This fitting of experimental data to a default model that however lacks a sound mechanistic justification, is already questionable in its own right. It introduces an element of subjectivity into data analysis, as it might not always be obvious which data points to exclude [31].

What is more, the information contained in the excluded data is lost. Bartlett and Gilbert argue this were of little relevance since the concentrations observed in vivo are much smaller than the concentration of maximal degradation [19]. While that might often be the case, it is not necessarily true for all PROTACs, especially for high-dose i.v. bolus administration. Under repeat dose regimens, drug accumulation may also lead to concentrations where the hook effect becomes relevant.

As we have demonstrated, the $E_{\max}$ model is also prone to overestimate the maximal extent of degradation ($D_{\max}$) for PROTACs that exhibit a pronounced hook effect (see Figure 3b). Therefore, rankings based on the $E_{\max}$ model tend to be biased in favor of such compounds which makes it harder to build reliable structure-activity relationships.

Crucially, the $E_{\max}$ model also cannot be used to assess whether observed data line up with expectations derived from PROTAC theory. In particular, no statement can be made about whether a drug’s behavior at concentrations above the $DC_{\max}$ is plausible. This makes it more difficult to identify assay artefacts and validate new experimental assays.

Finally, the best fit of the hook model can never perform worse than the best fit of the $E_{\max}$ model, as the two are nested (cf. Equations (12) and (13)). This even applies if there is no hook effect present in the data. Yet, to avoid overparameterization, the $E_{\max}$ model might still be preferred under such circumstances.

In summary, the hook model represents an improvement on the $E_{\max}$ model. It leads to better curve fits, provides more accurate parameter estimates and makes more sense mechanistically. An R-script for automated data analysis using the hook model is provided in the Appendix A.

#### 4.1.2. Guidance for Experimental Design

Empirical models describing in vitro target degradation as an exponential decay process have already been suggested elsewhere [32]. Here, we provide a mechanistic derivation that shows under which assumptions this kind of model would apply. We further investigated how the concentration-degradation profile depends on incubation time.

While the concentration of maximal degradation ($DC_{\max}$) stays constant, the maximal extent of degradation ($D_{\max}$) and especially the concentration of half-maximal degradation ($DC_{50}$) change significantly over time. This is an issue, since the indirect response models used for translating in vitro data to in vivo require the steady-state parameters as inputs [33,34,35,36,37]. If the $D_{\max}$ and $DC_{50}$ values for a standard incubation time (e.g., 24 h), are meaningfully different from their steady-state counterparts, those models will make biased inferences. In principle, two different strategies can be proposed to mitigate this risk:(I)Use of an incubation time that is long enough to directly yield the steady-state profile. However, the required incubation time will be different for different PROTAC concentrations [19]. Therefore, one cannot conclude that the steady state has been reached just because degradation at a particular concentration is the same over time. Instead, we propose a different approach that requires only rough estimates for protein half-life ($t_{½,P}$) and for the maximal extent of degradation ($D_{\max}$) as inputs. A threshold value for the minimal extent of degradation that is still acceptable is defined first (e.g., $D_{\max}\geq 80\%$). Next, the protein’s half-life or at least a rough estimate that should rather be too high than too low must be approximated. From these parameters, minimum required incubation times for in vitro degradation experiments can be calculated (see Table 2).(II)If target protein half-life is longer than 24 h, the theoretically required incubation times exceed what is practically feasible. For such cases, the extended hook model (Equation (14)) allows to estimate steady-state parameters from a pre-steady-state concentration-degradation profile obtained with standard incubation times. Yet, this second approach using the extended hook model (Equation (14)) is also more sensitive to measurement errors and requires precise estimates for protein half-life. These may be obtained by fitting concentration-degradation profiles at different incubation times in parallel.(Ⅲ)Finally, the question remains, which drug concentrations to study in vitro. To get the most reliable parameter estimates, one should cover the entire profile, including concentrations of no degradation up to concentrations displaying the hook effect. Demonstrating that there is a hook effect in protein degradation also is strong evidence in favor of the PROTAC mechanism of action. If no prior information about the relevant concentration range is available upfront, one could estimate $DC_{\max}$ from the binary binding affinities (see Appendix C) to get an initial idea. However, assay artefacts might occur more frequently at the highest concentrations, if for instance compounds become cytotoxic or poorly dissolved.


### 4.2. The $k_{cat}$ Model

#### 4.2.1. Lessons Learnt about the PROTAC Mechanism of Action

Two processes have previously been implied to be rate-limiting for PROTAC-mediated protein degradation. Those are ternary complex formation and monoubiquitination [18]. Here, we pursued a rapid equilibrium approach, assuming that monoubiquitination is the rate-limiting step. Since the resulting model predictions were accurate, this assumption seems justified. It is furthermore supported by previous reports in which monoubiquitination has been identified as the rate-limiting step for polyubiquitination [38].

Interestingly, the kinetics of complex formation appeared to be irrelevant here as they could be left out of the model altogether. This finding challenges the kinetic proofreading hypothesis, which proposes that ternary complexes with slower kinetics should be more efficient at mediating target degradation [8]. The idea being that only if a complex persists for long enough will the E3 ligase have enough time to catalyze polyubiquitination. However, this effect is only relevant if the subsequent ubiquitination reactions occur on the same time scale as monoubiquitination. At least for physiological polyubiquitination processes this does not seem to be the case [39].

It was an important finding that the rate constant of ubiquitination ($k_{\mathrm{cat}}$) was the same for different PROTACs, also across the different cells tested. Thus, target engagement (TE) and hence the three binding equilibrium constants ($K_{D,P}$, $K_{D,E}$ and α) determined which of the PROTACs led to the deepest target protein degradation. It remains to be seen, whether that also applies to compounds that are structurally more diverse than the ones from Zorba et al. [21]. Previous studies had already revealed that data on target engagement is not sufficient to accurately predict degradation across different target proteins [40]. This might either be due to the proteins having different baseline half-lives ($k_{\deg}$) or because they get ubiquitinated at different rates ($k_{\mathrm{cat}}$). In conclusion, $k_{\mathrm{cat}}$ may only depend on the target and the E3 ligase that ubiquitinates it, but not on the PROTAC that links the two.

If, following polyubiquitination, the ternary complex has to dissociate before degradation can occur [18], degradation can never be more rapid than dissociation itself. Thus, protein degradation is not expected to shift the thermodynamic binding equilibria, further justifying the rapid equilibrium assumption we made. This line of thinking also gives rise to the idea that ternary complexes that dissociate more slowly would give less degradation. However, our results do not support this notion of over stabilization [41], suggesting that ubiquitination alone is the rate-limiting step in protein degradation.

Finally, we noticed that already very little target engagement led to substantial protein degradation. This is due to the catalytic mechanism of action of PROTACs. The efficiency of catalysis is defined by the parameter $TE_{50}$ which gives the extent of target engagement required for 50% degradation. $TE_{50}$ also quantifies how much the PROTAC speeds up target degradation, as $TE_{50}$ is given by the ratio of $k_{\deg,P}$ and $k_{\mathrm{cat}}$ (Equation (15)). In the present example, PROTAC-mediated degradation was ~100 times faster than baseline turnover ($TE_{50}=k_{\deg,P}/k_{\mathrm{cat}}=0.043\;h^{-1}/4.6\;h^{-1}\approx 1\%$). This metric can be very useful to assess and compare PROTACs in early drug discovery.

#### 4.2.2. Model-Informed Drug Discovery

In the presented example, the extent of degradation was not only determined by the PROTAC’s affinity for the target protein ($K_{D,P}$) as the affinity for the E3 ligase ($K_{D,E}$) and the cooperativity factor (α) played an important role too. This is in line with previous studies that failed to find a correlation between $K_{D,P}$ and degradation [40]. Therefore, medicinal chemistry efforts should not be limited to only improving the drug’s affinity for the target protein. Instead, a more holistic approach, considering all three binding interactions is needed.

The $k_{\mathrm{cat}}$ model in turn allows to quantify how much the individual binding affinities would need to be increased to achieve the desired extent of degradation. Medicinal chemistry can then assess whether such improvements are feasible or whether a new compound series should be identified instead. In general, if target engagement is already much greater than $TE_{50}$, increasing binding affinities likely will not improve degradation. Instead, researchers should increase the rate of ubiquitination ($k_{\mathrm{cat}}$). This might be achieved by switching to another lead with different target and E3 ligase binding moieties or by recruiting a different E3 ligase.

When it comes to modeling ternary complex formation in biochemical assays, the concentrations of target protein ($P_{0}$) and targeted E3 ligase ($E_{0}$) are key parameters [6,7]. However, only few attempts have been made at applying this insight to target degradation and even those were limited to qualitative considerations [42]. Here, we used our $k_{\mathrm{cat}}$ model to quantitatively translate target degradation from one cellular system to another based on the expression levels of E3 ligase. This approach can be useful when the cell type of interest cannot be studied in vitro. Degradation could then be assessed in related cells and the results can then be translated back to the cell type of interest.

Target protein expression levels, in contrast, seemed to play only a minor role here. In theory, they should only affect degradation if they are high enough to saturate the E3 ligase. In all other cases, only the levels of E3 ligase in different cell lines but not those of the target protein need to be determined. This reduces experimental efforts, since the targeted E3 ligases are often the same for different projects [43].

In contrast to the target protein’s baseline expression levels, its baseline half-life ($t_{½,P}$) was found to be perhaps the single most critical physiological parameter. It not only influences the extent of degradation, but also determines the kinetics of degradation. Therefore, even if degradation data on the relevant cell type is available, it still takes $t_{½,P}$ to predict the time-course of degradation and target recovery. This is not limited to the in vitro scenario, but it also applies to the indirect response models used for in vivo modeling [33,34,35,36,37]. There too, degradation introduces a substantial time-delay between exposure and a drop in protein levels, which must be considered by the model [23]. Thus, we strongly suggest determining the target protein’s half-life early on in the drug discovery project. There are several methods available for doing so, such as SILAC [24] or the cycloheximide chase assay [44].

### 4.3. The PD Model

#### 4.3.1. Leveraging the Inhibitory Activity of PROTACs

In order to make predictions about a downstream pharmacodynamic effect, a relationship between that effect and target degradation has to be established. This also applies to the in vivo scenario, where simple empirical models have already been used successfully to translate protein degradation to cytokine response [45]. Our PD model provides a generalized version of this approach and in addition also incorporates the effect of target inhibition by the PROTAC. Theses inhibitory effects are particularly relevant at high concentrations and at early time points.

Although target degradation is frequently associated with a hook effect at high concentrations [27,28,29,30], the same may not be true for downstream pharmacodynamic effects. There, target inhibition by the PROTAC is expected to compensate for the loss in degradation resulting from the hook effect. Therefore, the hook effect should be of only little concern for pharmacodynamic effects in vivo, provided the target can also be inhibited by the PROTAC. This benefit, however, is not available for non-inhibiting PROTACs, e.g., those targeting scaffolding proteins [46], where a hook effect in target degradation may also translate to reduced downstream pharmacodynamic effects.

While there might be a relevant time delay between drug administration and target degradation, inhibition acts more rapidly. Thus, the additional inhibitory activity of PROTACs might lead to an even more sustained pharmacodynamic response in vivo. Inhibition would allow for a rapid onset of the effect, whereas degradation would ensure its persistence. As a result, lower doses might be feasible for PROTACs compared to conventional target inhibitors.

#### 4.3.2. Model-Informed Target Validation

It is critical for the rational design of PROTACs to demonstrate early on that it is really degradation of the intended target that causes the pharmacodynamic response. This can be done in a two-step approach with the help of the PD model.

First, comparing the PROTAC to a non-degrading control compound allows to deconvolute effects that are due to inhibition from those caused by degradation. In principle, a PROTAC can act (1) only as a degrader, (2) only as an inhibitor or (3) as both, a degrader and an inhibitor. The PD model resolves this uncertainty by quantifying the relative contributions of degradation and inhibition to the overall PROTAC effect (see Figure 6b,d).

Second, comparing several PROTACs from the same project helps to exclude the involvement of off-targets in the PD effect. The inhibitory potencies of the respective degradation-incompetent controls ($IC_{50}$) are used to quantify the effects that are due to target inhibition. The remaining pharmacodynamic response is then attributed to degradation of the target protein. If the pharmacodynamic effect produced by a certain extent of degradation is the same among the PROTACs this is indicative of on-target effects. If instead the same extent of degradation gives different PD responses for the PROTACs, off-target degradation must be considered.

## 5. Conclusions

The presented modeling framework has three principal applications in the early phases of the drug discovery of PROTACs (see Figure 7).

First, the *hook model* (Equation (12)) allows to describe the PROTAC concentration-degradation profile (see Figure 7a). It takes experimental data on target protein degradation as input and returns three fitted parameters ($D_{\max}$, $DC_{50}$ and $DC_{\max}$), which can be used to assess, rank and compare different PROTACs. The extended hook model (Equation (14)) can be used to estimate the steady-state parameter values by extrapolating in time, provided that information on the target protein’s half-life ($t_{½,P}$) is available.

Second, the $k_{cat}$
*model* (Equation (11)) guides the rational optimization of a PROTAC lead compound (see Figure 7b). As input, it requires biochemical data on the PROTAC’s binding affinities ($K_{D,P}$, $K_{D,E}$ and α) as well as information on the relevant cellular system ($E_{0}$ and $t_{½,P}$). The pivotal parameter $k_{\mathrm{cat}}$ is obtained by fitting the concentration-degradation profile for the lead compound. Based on that value, the model then allows to predict what improvements in the binding affinities are necessary to get the desired level of target degradation. The model also allows to translate results between different cell types.

Third, the PD
*model* (Equation (20)) informs pharmacology by predicting a target value for protein degradation to achieve a desired level of a downstream PD effect (see Figure 7c). In addition to the three degradation parameters ($D_{\max}$, $DC_{50}$ and $DC_{\max}$), the PD model also considers, if relevant, the PROTAC’s inhibitory potency ($IC_{50}$). The parameter that links degradation to downstream pharmacodynamic effects ($P_{50}$) is informed by fitting the model to data obtained from the lead compound. The resulting model can then predict which improvements in the degradation parameters are necessary to achieve the desired pharmacodynamic effect.

Taken together, the presented modeling framework provides a comprehensive and very practical toolbox that enables model-informed assessment and optimization of PROTACs in the early phases of drug discovery using in vitro degradation and pharmacodynamic data. An extension of this framework towards in vivo and translational aspects is in progress.

## Figures and Tables

**Figure 1 pharmaceutics-15-00195-f001:**
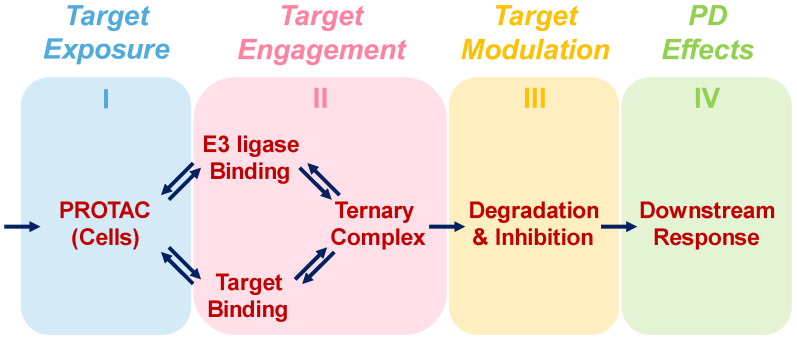
The concept of the four pillars of translational pharmacology is applied to PROTACs. First, the PROTAC has to get access to its site of action. Next, it must bind to both the target protein as well as the targeted E3 ligase to form a ternary complex. That ternary complex must then mark the target protein for degradation by the proteasome. Finally, degradation of the target protein together with inhibition of what activity remains must translate to the relevant downstream effect.

**Figure 2 pharmaceutics-15-00195-f002:**
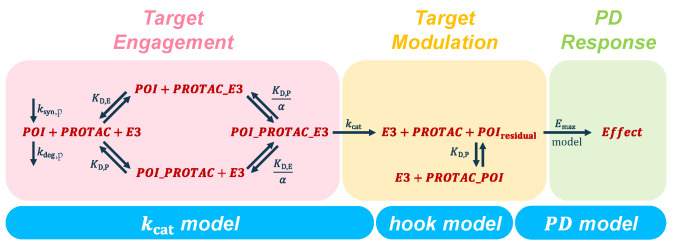
The pharmacodynamic modeling framework for PROTAC assessment and characterization consists of three models. (1) The $k_{cat}$
*model* links the biochemical parameters which govern target engagement to target degradation. (2) The *hook model* relates target degradation to PROTAC concentrations and incubation time. (3) The PD
*model* integrates two mechanisms of target modulation, i.e., target degradation and target inhibition, and translates them to a downstream pharmacodynamic readout.

**Figure 3 pharmaceutics-15-00195-f003:**
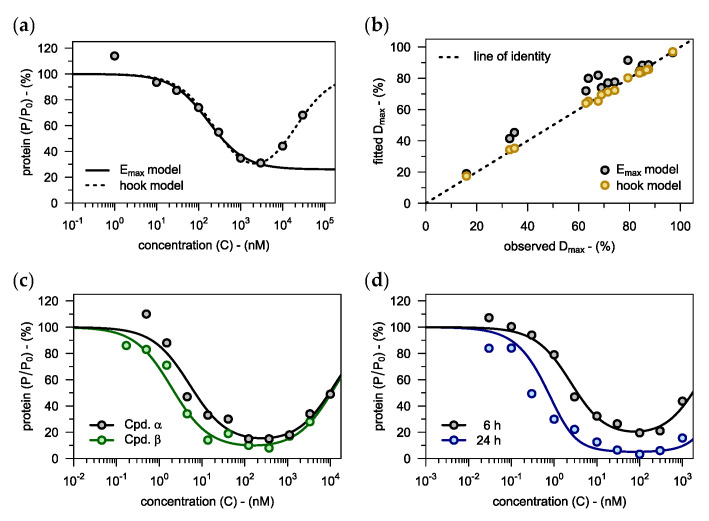
(**a**) Relative levels of target protein are plotted against PROTAC concentration in media (i.e., concentration-degradation profile). For comparison, both the hook model presented here as well as the conventional $E_{\max}$ model are fitted to the data (see Appendix A for further examples). When fitting the $E_{\max}$ model, only concentrations below the concentration of maximal degradation were considered. (**b**) A total of 17 concentration-degradation profiles were described using both the hook model and the $E_{\max}$ model. The resulting estimates for the maximal extent of degradation ($D_{\max}$) are plotted against the respective experimentally observed values. Experimental data (**a**,**b**) taken from Zorba et al. [21]. (**c**) The hook model is applied to reversible covalent (*Cpd. α*) and irreversible covalent PROTACs (*Cpd. β*). Data taken from Gabizon et al. [22]. (**d**) Protein levels after 6 h of incubation as well as levels after 24 h are plotted against the concentration of the non-covalent PROTAC *Cpd. Y*. The extended hook model (Equation (14)) is fitted to the concentration-degradation data from the 6 h time point, which then allows to predict degradation after 24 h. The experimental observations for the 24 h time point confirm that prediction. Data taken from Mares et al. [23].

**Figure 4 pharmaceutics-15-00195-f004:**
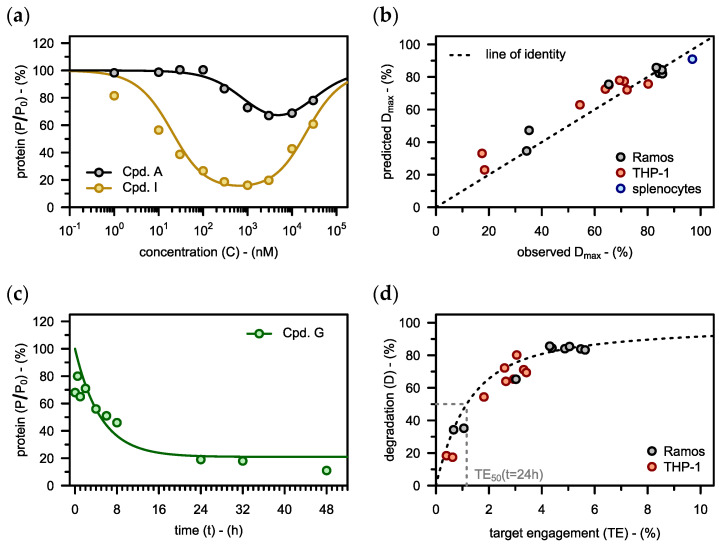
(**a**) The rate constant of PROTAC-catalyzed protein degradation is fitted to the concentration-degradation profile observed for the lead compound (*Cpd. A*) in Ramos cells ($k_{\mathrm{cat}}=4.6\;h^{-1}$). This is the only parameter in this figure that was derived by model fitting. Subsequently, the model is used to predict, what binding affinities would give the orange profile. The optimized compound (*Cpd. I*), which features the binding affinities suggested by the model (see Appendix A), exhibits the desired concentration-degradation profile. (**b**) Based on the $k_{\mathrm{cat}}$ value from *Cpd. A* and the biochemical input data (Appendix A), the maximal extent of degradation ($D_{\max}$) is predicted for different compounds in different cell types. These predictions are plotted against the estimates for $D_{\max}$ obtained by characterizing the respective experimental concentration-degradation profiles with the hook model (see Figure 3b). There is good agreement between the predicted and the observed values, as indicated by Pearson’s correlation coefficient (ρ=0.97). (**c**) The time-course of degradation in Ramos cells following incubation with *Cpd. G* (C=100 nM) is predicted using the $k_{\mathrm{cat}}$ value from *Cpd. A*. (**d**) The maximal extent of degradation ($D_{\max}$) observed for the nine compounds from before is plotted against the respective target engagement (TE) values calculated using Equation (3). Equation (15) predicts a hyperbolic relationship ($TE_{50}\approx 1\%$). Experimental data (**a**–**d**) taken from Zorba et al. [21].

**Figure 5 pharmaceutics-15-00195-f005:**
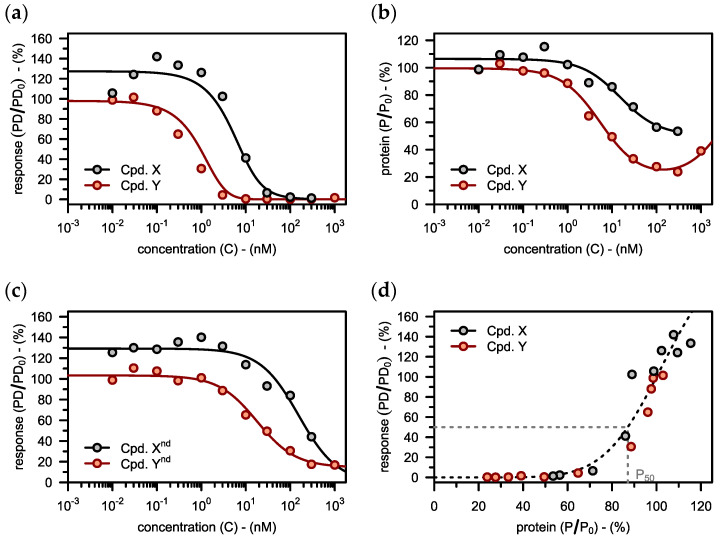
(**a**) The PD model is fitted to data on the downstream pharmacodynamic response (TNF-alpha levels) in the presence of various concentrations of the lead compound (*Cpd. X*). This analysis yields the target protein levels of half-maximal pharmacodynamic response ($P_{50}=87\%$) and an empirical hill coefficient (n=7). Subsequently, a target profile is defined for a potential follow-up compound (*Cpd. Y*). (**b**) Target protein levels (RIPK2) are plotted against drug concentration. The PD model is used to predict the target concentration-degradation profile for the follow-up compound from before (*Cpd. Y*). (**c**) The downstream pharmacodynamic response (TNF-alpha levels) is plotted against the concentration of the corresponding non-degrading (nd) control compounds. (**d**) The observed TNF-alpha response is plotted against the corresponding RIPK2 levels for both compounds. The dashed line shows the fitted PD model from before, but this time explicitly neglecting inhibition (I=0). Experimental data (**a**–**d**) taken from Mares et al. [23].

**Figure 6 pharmaceutics-15-00195-f006:**
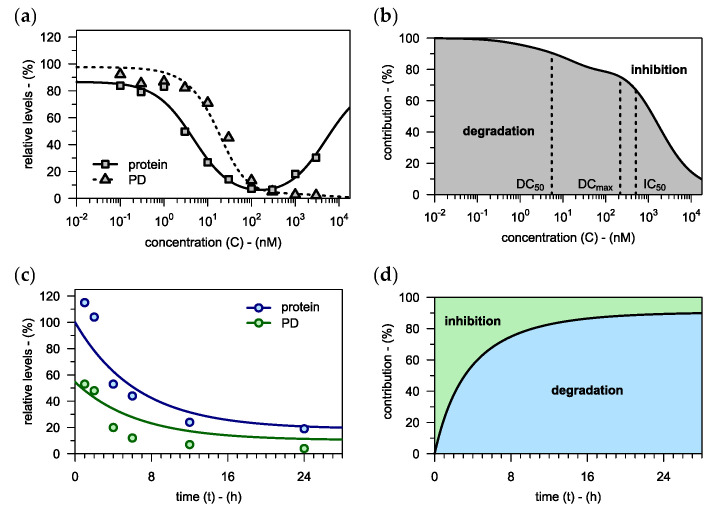
(**a**) Target protein levels and the downstream pharmacodynamic response are plotted against drug concentration for an in-house PROTAC (see Appendix A for two more examples). The hook model is used to assess degradation, which is then used to fit the PD model (fitted to all three compounds simultaneously). As predicted by the model (see Appendix E for derivation), there is no hook effect present on the level of the downstream pharmacodynamic response. (**b**) The relative contributions of target degradation and target inhibition to the overall pharmacodynamic effect are plotted against PROTAC concentration. At higher drug concentrations, inhibition becomes the dominant driver of pharmacodynamic effects, thus compensating for the hook effect in protein levels. (**c**) Target protein levels and the downstream pharmacodynamic response are plotted against incubation time for an in-house PROTAC which was applied at a constant concentration of C=100 nM. The time-course of degradation is predicted using the extended hook model (Equation (14)). Based on this prediction and considering the compound’s inhibitory potency, the PD model is used to also predict the time-course of the pharmacodynamic response. Both predictions are in good agreement with the observed data. (**d**) The relative contributions of target degradation and target inhibition to the overall pharmacodynamic effect are plotted against incubation time. Initially, the downstream pharmacodynamic effect is dominated by inhibition, but over time, degradation becomes more important.

**Figure 7 pharmaceutics-15-00195-f007:**
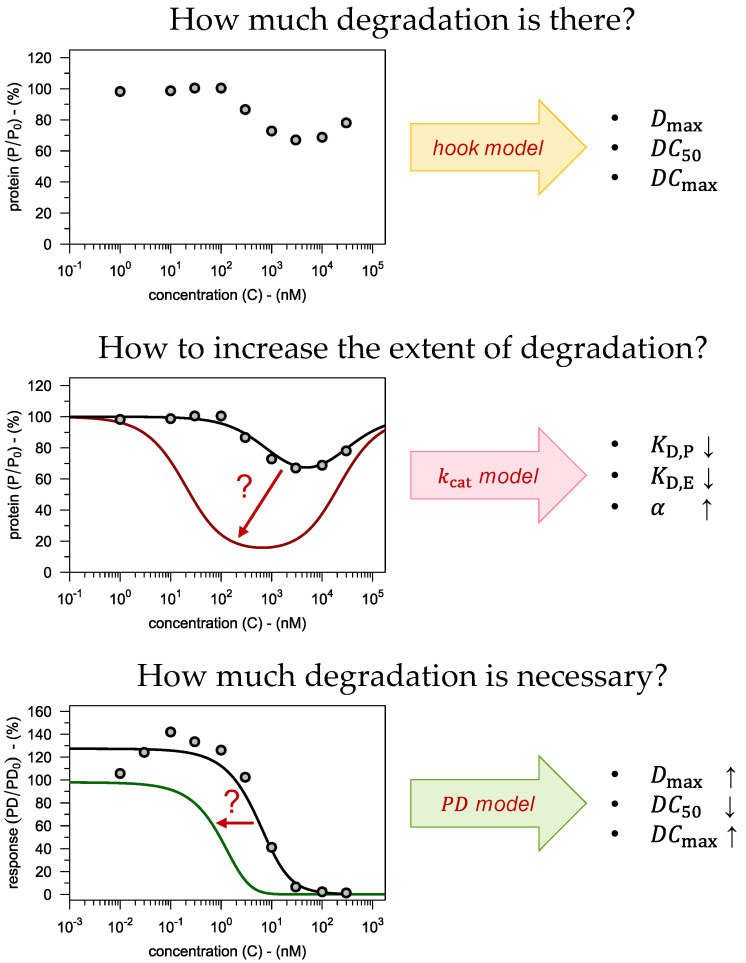
Our modeling framework can be used to address three major questions, each associated with one of the pillars of translational pharmacology (see Figure 1 and Figure 2 for color coding). The hook model is fitted to the concentration-degradation profile yielding three parameters which can be used to assess PROTACs as degraders. The $k_{\mathrm{cat}}$ model allows to predict what binding affinities are necessary to achieve the desired degradation, thereby giving rational guidance to lead optimization efforts. The PD model translates protein degradation to a defined biomarker response. This is particularly useful when it comes to identifying a target value for protein degradation.

**Table 1 pharmaceutics-15-00195-t001:** The descriptive parameters corresponding to the two incubation times are stated together with their steady-state counterparts. Parameter values for 6 h and 24 h incubation were obtained by fitting two individual hook models (Equation (12)) to the data. Fitting the extended hook model (Equation (14)) to both time points simultaneously allows to estimate the steady-state values. Least squares estimates are reported with the upper and lower bounds of the bootstrap 95% confidence intervals given in brackets. An asterisk (*) indicates that a parameter is significantly different from its steady-state counterpart, according to Welch’s *t*-test. Experimental data taken from Mares et al. [23].

t (h)	$D_{\max}\;\left(\%\right)$	$DC_{50}\;\left(\mathrm{nM}\right)$	$DC_{\max}\;\left(\mathrm{nM}\right)$
6	79.8 [73.5, 84.2] *	2.19 [1.41, 5.15] *	79.8 [59.5, 288]
24	95.5 [91.5, 97.2]	0.65 [0.32, 1.46] *	73.7 [48.1, 142]
Steady State	94.9 [93.5, 95.9]	0.29 [0.15, 0.66]	68.9 [47.1, 104]

**Table 2 pharmaceutics-15-00195-t002:** The minimum incubation time (h) necessary to get accurate estimates for $D_{\max}$ and $DC_{50}$ (i.e., estimates that are less than a factor of 1.5 ($DC_{50}$) or less than 5% ($D_{\max}$) different from the steady-state values) is given for different threshold values ($D_{\max}\geq x$) and different proteins ($t_{\text{½},P}\leq x$). A green box indicates that 24 h should be sufficient, while orange means that it takes longer than that. The required incubation time decreases with increasing $D_{\max}$ and decreasing $t_{\text{½},P}$.

	$D_{\max}\;\left(\%\right)$									
$t_{½,\;P}\;\left(h\right)$	30	50	60	70	75	80	85	90	95	99
**2**	5	4	4	3	3	3	3	3	3	3
**4**	9	8	7	6	6	6	6	5	5	5
**6**	13	11	10	9	9	9	8	8	7	7
**12**	26	22	20	18	18	17	16	15	14	13
**24**	51	44	40	36	35	33	31	29	27	26
**36**	77	66	60	54	52	49	46	43	41	38
**48**	102	87	80	72	69	65	61	58	54	51
**72**	153	131	120	108	103	97	92	86	81	76
**96**	204	174	159	144	137	130	122	115	107	101
**192**	407	348	318	288	273	259	244	229	214	202

## Data Availability

Data were taken from publications (see manuscript for details).

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
