# Peer review of "A Mechanistic Pharmacodynamic Modeling Framework for the Assessment and Optimization of Proteolysis Targeting Chimeras (PROTACs)"

_pharmaceutics, 2023, doi:10.3390/pharmaceutics15010195_

Round 1

Reviewer 1 Report

I find it very difficult for me to follow this manuscript likely due to 1) incomplete/missing derivation of the model equation 2) various data sources (at least three different ones I found) including in-house and multiple papers. The author did not include these data in the method section 3) missing link between the method and results. It is not clear what questions the authors are trying to address and how each method was used to obtain results. Please provide your data source in the material and method section; and link these data with the model you used; and explain what are the questions you are trying to address in each result section. It seems to me that data and models were used for three later steps in Figure 1. However, I cannot connect them. I also have the following specific question

1.       Please provide definitions Dmax and Dc50 in line 66 in the introduction  

2.       I cannot find equation A4 in appendix in the reference 6. TC corresponds to “PLE” in Han et al. Please provide details location of equation A4.

3.       Why TC<<E0 was assumed in equation A2. Similarly, can you assume TC<<P0?

4.       It is not clear how equation 12 was derived. Not able to drive equation 11 to equation 12. Please provide detailed derivation.

5.       Please provide what is Dmax and how it is derived as shown in A7 Appendix B

6.       It is also not clear how A8 and A9 in the appendix were derived.

7.       Why the target degradation was called kcat model or hook model, what is the difference?

8.       Equation 14 seems to be similar to equation 11 except for Dss(C) definition why

9.       It is not clear why equation 12 can be approximated by equation 13.

10.   Please show how equation 15 was derived

11.   It is not clear how the pillars in materials and methods were applied in the result section especially pillar III.b and pillar IV

12.   Missing R-script for data analysis

13.   Table 1, usually the capacity parameter and efficacy parameter should not change over time. This suggests that some assumptions should not be made. On page 13 author made some methods to mitigate the risk. I do not understand why they are time-dependent. If so, why cannot use a model with ODE to consider the time?

Author Response

Thank you very much for your diligent review. We highly appreciate your comments, especially those dealing with how we could make the Methods section more comprehensible. Your remarks prompted us to convey the main messages of our work more clearly. We achieved this by 1) providing an additional subchapter at the end of Methods, where we link the individual models to the pillars of translational pharmacology and to the questions they address and by 2) greatly expanding on the level of detail and the number of intermediate steps in our mathematical model derivations. Therefore, we think that your contribution greatly increased the qualiy of our manuscript.

Please see the attachment for our detailed point-by-point response.

Reviewer 2 Report

Andreas Reichel and a co-worker report the framework for a mechanistic pharmacodynamic modeling for the assessment and optimization of PROTACs. The PROTACs has been proven to be effective method in drug discovery, and this work aim to establish pharmacokinetic/pharmacodynamic models of PROTACs. This work developed a modeling framework that can be applied to experimental data from routine projects, involving three processes: 1) assess PROTACs  based on accurate degradation metrics, 2) guide compound optimization of the most critical parameters, and 3) link degradation to downstream pharmacodynamic effects.

This manuscript is well written, and it could be useful for future investigators in the field of PROTACs.

The comments and suggestions are listed below.

1.     Keywords should contain the word “Proteolysis targeting chimeras”.

2.     Figure 1.; the number I, II, III, and IV  should be without parenthesis.   I), II), II)…..  in the Figure 1 are difficult to read.

3.     Please revise “However, what can be measured instead is the effect the corresponding non-degrading inhibitor has on the pharmacodynamic response.”  to However, what can be measured instead is the effect the corresponding non-degrading inhibitor having on the pharmacodynamic response.”.

4.     “Importantly, all three of these parameters need to be considered, indicating that no single binding interaction alone is key.”. Please consider “Importantly, all three of these parameters need to be considered, indicating that no single binding interaction alone is the key parameter.”.?

Author Response

Thank you very much for your positive feedback and for your valuable inputs.

Please see the attachment for our detailed point-by-point response.

Reviewer 3 Report

The article “A Mechanistic Pharmacodynamic Modeling Framework for the Assessment and Optimization of PROTACs” is very interesting but I have the following comments and suggestions,

1. To increase the interest of readers it is suggested that the abbreviation PROTACs should be replaced with full text in the title.

2. The abstract contains no information regarding the results.

3. The Introduction section is well written.

4. The method section is very well written, and all the relevant information has been provided efficiently.

5. The graphs and figures are very informative. 

Author Response

Thank you very much for your positive feedback and for your valuable inputs. Your remarks about the Abstract prompted us to elaborate more on the main findings of our work there, which greatly increased the quality of our manuscript.

Please see the attachment for our detailed point-by-point response.

Reviewer 4 Report

After reading the manuscript in detail, I suppose that the all paragraphs together describe the presented study in broad and appropriate way. The conclusions presented by the authors are consistent with the evidence and relate to the main research issue. However, I recommend to extend the validation methods. In chapter 2.1 authors described, " The model was built and validated using in-house and literature data". In chapter 3.3.1 the experimental data were taken only from Mares et al. [23] article and in-house data are missing. Moreover, in figures S1 (all) and S2 (d-h), within the picture are two solid lines. According to explanation "hook-model" line should be dashed.

Author Response

(The authors gave the same response as above.)
